# Control of biosilica morphology and mechanical performance by the conserved diatom gene *Silicanin-1*

Stefan Görlich[1], Damian Pawolski[1], Igor Zlotnikov[1] & Nils Kröger [1]

The species-specifically patterned biosilica cell walls of diatoms are paradigms for biological mineral morphogenesis and the evolution of lightweight materials with exceptional mechanical performance. Biosilica formation is a membrane-mediated process that occurs in intracellular compartments, termed silica deposition vesicles (SDVs). Silicanin-1 (Sin1) is a highly conserved protein of the SDV membrane, but its role in biosilica formation has remained elusive. Here we generate *Sin1* knockout mutants of the diatom *Thalassiosira pseudonana*. Although the mutants grow normally, they exhibit reduced biosilica content and morphological aberrations, which drastically compromise the strength and stiffness of their cell walls. These results identify *Sin1* as essential for the biogenesis of mechanically robust diatom cell walls, thus providing an explanation for the conservation of this gene throughout the diatom realm. This insight paves the way for genetic engineering of silica architectures with desired structures and mechanical performance.

---

[1] B CUBE Center of Molecular Bioengineering, CMCB, TU Dresden, Am Tatzberg 41, 01307 Dresden, Germany. Correspondence and requests for materials should be addressed to N.K. (email: nils.kroeger@tu-dresden.de)

Diatoms are a large group of single-celled microalgae (~15,000 described and >100,000 estimated species) renowned for their silica-based cell walls that exhibit species specifically ornamented morphologies with patterns of nano- to micrometer-sized pores[1]. Due to this remarkable structural versatility, diatom biosilica formation is a paradigm for studying the mechanisms of genetically controlled mineral morphogenesis[2–4]. It has been proposed that the ecological success of diatoms is in part due to their capability of producing biosilica cell walls, which may serve as an efficient armor against grazers and parasites[5,6]. Indeed, diatom biosilica exhibits the highest specific strength of any known biological material[7]. Understanding the correlation between biosilica morphology and its mechanical properties is therefore also of great interest for engineering light-weight materials with high mechanical stability[7,8]. Previous biochemical analyses and transcriptomics studies have identified many proteins that are hypothesized to be involved in biosilica formation[4,9–12]. The recent establishment of gene knockdown and knockout methods for diatoms[13,14] has provided the ability to investigate the role of these proteins in biosilica morphogenesis in vivo. Two recent studies have reported siRNA- and RNAi-mediated knockdown of the genes SAP1, SAP3, and silacidin that code for biosilica-associated proteins in the diatom Thalassiosira pseudonana[15,16]. Silacidin knockdown mutants have an approximately twofold bigger cell volume, but exhibit the same biosilica morphology as the wild type[15]. This result has been confirmed through knockout of the silacidin gene[17]. In contrast, the SAP1 and SAP3 knockdown mutants display aberrations in biosilica morphology[16]. However, uncertainties remain as potential off-target effects were not investigated, and it was not addressed whether the amount of SAP1 and SAP3 downregulation in the mutant strains correlated with the extent of morphological change. Recently, a new biosilica-associated protein, Silicanin-1 (Sin1), that is highly conserved in diatoms was discovered[18]. Sin1 is a 45 kDa type-1 membrane protein (Supplementary Fig. 1A) located in silica deposition vesicles (SDVs)[18]. SDVs are the lipid bilayer bounded intracellular compartments in which biosilica formation and morphogenesis takes place[19]. In combination with long-chain polyamines, clusters of recombinant Sin1 molecules strongly promote silica formation from silicic acid in vitro[18]. It was therefore hypothesized that Sin1 is involved in the biosynthesis of diatom biosilica[18]. Here we investigated this hypothesis through the knockout of the Sin1 gene using a CRISPR/Cas9-based approach. The Sin1 knockout mutant was viable, yet it exhibited significant deficiencies in the formation of biosilica both in quantity and in quality, which resulted in a phenotype with decreased mechanical stability.

## Results

### Inactivation of the Sin1 gene using CRISPR-Cas9. 
The DNA construct for knocking out the Sin1 gene in T. pseudonana, pSin1KO (Supplementary Fig. 1B), was analogous to a recently described plasmid for knocking out the urease gene in the same organism[20]. Two DNA sequences encoding Sin1 specific guide RNAs (gRNA-1 and -2) were each placed under the control of the T. pseudonana U6 promoter[20]. gRNA-1 was designed to target the most 5′-located region of Sin1, while gRNA-2 targeted a region 411 bp further downstream (Supplementary Fig. 1A). Plasmid pSin1KO was introduced into T. pseudonana by microparticle bombardment and nourseothricin-resistant clones were selected on agar plates[21]. Individual clones were grown in liquid medium, and 8 of 18 transformants clones exhibited Cas9-GFP fluorescence in the nucleus. Sequencing of Sin1-specific genomic PCR products for three of these clones (C4, C5, and C6)

revealed numerous sequence ambiguities at the predicted Cas9-induced double-strand break site that was targeted by gRNA-1 (Supplementary Fig. 2). This suggested that these primary clones were mosaic, containing a mixture of wild type and mutant Sin1 genes. Mosaic colonies have previously been observed in diatoms that were subjected to CRISPR/Cas9-based genome editing[20,22]. Therefore, each of the three clones was subcloned twice by re-plating, and a total of 48 subclones from the third generation were analyzed by sequencing of the Sin1-specific genomic PCR products. The PCR products from subclones C4-15-1 and C6-6-1 yielded unambiguous sequences (Supplementary Fig. 3A) lacking 2 bp (positions 37 and 38) and 16 bp (positions 34–49), respectively (Supplementary Fig. 3B). For simplicity, sub-clones C4-15-1, C5-21-8, and C6-6-1 were re-named knockout-1, knockout-2, and knockout-3, respectively. The PCR product from sub-clone knockout-2 contained two different bases at many nucleotide positions around the predicted double-strand break site (Supplementary Fig. 3A). This sequence ambiguity was due to the presence of two different mutated Sin1 alleles, one containing a 4-bp deletion (positions 34–37) and the other a 25-bp deletion (position 27–51; Supplementary Fig. 3B). The sequence analysis demonstrated that each of the three third-generation sub-clones carried only mutated Sin1 alleles with frameshift deletions. The mutated Sin1 alleles encode drastically shortened polypeptides (78–96 amino acids) that have only the first 9–12 amino acids in common with the wild-type Sin1 protein (426 amino acids). Therefore, each of the three sub-clones lacks a functional Sin1 gene and thus represents a knockout mutant. Throughout our screening procedure no clones were obtained that exhibited Sin1 mutations in the region targeted by gRNA-2, suggesting that this guide RNA was inactive.

To investigate whether other parts of the T. pseudonana genome besides the Sin1 gene have been affected in the knockout mutants, we analyzed potential off-target sites for gRNA-1 and -2. The online software tool Cas-Offinder[23] predicted that no binding sites with less than four mismatches for either gRNA-1 or gRNA-2 are present in the T. pseudonana genome (Supplementary Table 1). With four mismatches allowed, there are four additional binding sites for gRNA-1 but still no binding site for gRNA-2 other than to the Sin1 gene (Supplementary Table 1). From genomic DNA of clone knockout-1 about 1 kb genomic DNA fragments around each of the four potential off-target sites for gRNA-1 were amplified by PCR. All four PCR products exhibited identical sequences to the same sites amplified from genomic DNA of the wild type (Supplementary Fig. 4), demonstrating that these sites were not affected by Cas9. This result is in line with data from a previous study, which demonstrated that potential off-target sites with only three mismatches were not affected by Cas9 in the diatom Phaeodactylum tricornutum[24].

The Cas9 gene had been placed under the nitrate reductase promoter, because we initially intended to permanently switch off Cas9 expression after the knockout mutation in the Sin1 gene had been introduced. However, growing T. pseudonana wild-type cells in $NH_4^+$-bearing culture medium led to a minor yet non-negligible fraction of cells with aberrant morphology. The morphological aberrations vanished completely within 2 weeks after the cells were transferred into $NO_3^-$-bearing medium. Since the morphological stability of the wild-type cells was an important reference point for the characterization of our knockout mutants (see below), we abandoned the original plan of switching off Cas9 and kept growing the mutant cells in $NO_3^-$-bearing medium. All following phenotype analyses were conducted with wild-type and mutant cell lines that were grown for >3 months in $NO_3^-$-bearing medium.

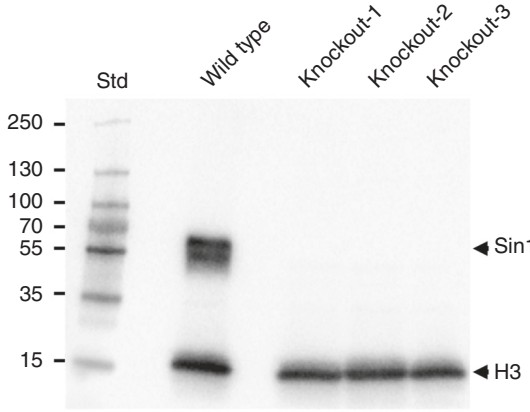

**Fig. 1** Western blot analysis for the presence of the Sin1 protein in *T. pseudonana* wild-type and mutant cell lines. Total lysates from equal amounts of cells were loaded in each lane. The blot was simultaneously probed with anti-Sin1 antiserum[18] and anti-Hitstone-3 antibodies. Lane Std was loaded with standard proteins with the indicated molecular masses

| Table 1 Growth rates and cellular silica content of *T. pseudonana* wild-type and three Sin1 knockout mutants | | |
| --- | --- | --- |
| **Strain** | **Growth rate ($10^5$ cells mL$^{-1}$ d$^{-1}$)** | **Silica content (fmol Si cell$^{-1}$)** |
| Wild type | 1.8 ± 0.2 | 110.7 ± 12.9 |
| Knockout-1 | 1.9 ± 0.5 | 72.0 ± 11.0 |
| Knockout-2 | 2.2 ± 0.2 | 75.8 ± 6.6 |
| Krockout-3 | 2.3 ± 0.3 | 74.0 ± 6.9 |

The growth rate for each strain was calculated from three independent biological replicates. Cells were grown for 19 days (Supplementary Fig. 5). Silica content and cell sizes (Supplement Table 2) were determined at around -$10^6$ cells mL$^{-1}$, which was within the linear growth regime (Supplementary Fig. 5)

**Physiological properties of the knockout mutants**. To validate the absence of the Sin1 protein in the knockout mutants, western blot analysis with total cell lysates was performed using a previously described anti-Sin1 antiserum[18]. As a loading control, the blot was simultaneously probed with antibodies against Histone-3 (15.3 kDa predicted molecular mass). As expected, the wild-type cells contained a Sin1 protein band (55 kDa apparent molecular mass)[18], whereas this protein was entirely missing in the cells from the three *Sin1* knockout strains (Fig. 1, Supplementary Fig. 5). The lack of Sin1 protein did not significantly change the growth rates of the knockout mutants (Table 1, Supplementary Fig. 6), indicating that the general metabolism required for protoplast proliferation was not affected. However, the total cellular silica content of *Sin1* knockout clones was only about two-thirds of the silica content in wild-type cells (Table 1). Taking into account slight differences in cell sizes, the silica content per cell wall surface area in the mutants was between 67–77% of the value for the wild type (Supplementary Table 2). This demonstrated that the Sin1 protein plays a role in the biogenesis of diatom biosilica.

**Analysis of biosilica morphology**. Diatom biosilica is composed of two different building blocks: valves, which are produced during cell division, and girdle bands, which are produced during interphase. The biosilica cell wall of *T. pseudonana* is cylindrical with one valve at the top and one at the bottom and multiple overlapping girdle bands connecting the two valves. The girdle band region of the wild type exhibits a characteristic pattern of porous and non-porous regions (Supplementary Fig. 7A). Girdle

bands that are close to the valves exhibit a higher density of pores than girdle bands located in the mid region of the cylinder. The girdle band biosilica of the three knockout mutants shows the same pattern of porous and non-porous regions, and the same density gradient of pores as the wild type (Supplementary Fig. 7B–D). In contrast, the silica patterns of the valves in the three knockout mutants exhibited differences compared to the wild type (Fig. 2), which is detailed in the following. The silica pattern in the wild-type valves is characterized by ribs of silica-termed costae that emanate from close to the valve center and branch out towards the rim of the valve (green lines in Fig. 2a). Neighboring costae are connected by seemingly irregularly spaced short silica bridges (orange lines in Fig. 2b), which in the following will be referred to as cross-connections. Two neighboring cross-connections together with the interjacent segments of the neighboring costae constitute an areola pore, which has a trapezoid shape (red line in Fig. 2a). An areola pore usually encloses several small pores that are 18.3 ± 3.1 nm in diameter[25] and are termed cribrum pores (Fig. 2a). Multiple tube-like structures, termed fultoportulae, are regularly spaced near the rim of the valve (yellow circles in Fig. 2a). Half of the valves contain one or two fultoportulae also near the center (blue circle in Fig. 2a). In the knockout clones, the circular shape of the valves, the number and arrangement of fultoportulae, and the dendritic patterns of costae were unchanged. However, the cross-connections were largely absent. As a consequence, the knockout clones lacked almost entirely the pattern of areolae pores (Fig. 2b–d).

As the number of cross-connections per valve showed a large variation, especially in the wild type, we manually counted them in 20 TEM images each of the wild-type and the three knockout mutants. To eliminate the influence of valve diameter, the density of cross-connections per square area was calculated for each valve (Fig. 3a). The result confirmed that each of the three knockout mutants contained a significantly lower density of cross-connections than the wild type (Fig. 3a). The median cross-connection density for the wild type (10.6 µm$^{-2}$) was 4–13-fold higher than for the knockout mutants (knockout-1: 0.9 µm$^{-2}$, knockout-2: 2.6 µm$^{-2}$, and knockout-3: 0.8 µm$^{-2}$). In the *T. pseudonana* wild-type valves costae are elevated in the distal direction (z-height) relative to the silicified region between the costae (i.e., the cribrum plate)[25]. In SEM images the costae had a flatter appearance than in the wild type (Supplementary Fig. 7). This was confirmed by atomic force microscopy (AFM) measurements with biosilica valves (Supplementary Fig. 8), which yielded 14.8 nm as the median z-height of costae for the wild type and 8.8 nm for the KO1 mutant (Fig. 3b).

**Characterization of the mechanical performance**. We hypothesized that the altered valve structures as well as the reduced silica content of the *Sin1* knockout strains should have a notable effect on the mechanical behavior of their biosilica cell walls compared to the cell walls from wild type. This was investigated by indenting the valve of isolated cell walls using a cube-corner diamond tip inside an electron microscope in two experimental modes. One was a displacement controlled mode with a maximum penetration depth of 1 µm (Fig. 4a–f), and the other was a load-controlled mode with a maximum force of 10 µN (Fig. 4g). The relatively sharp tip was used to concentrate the stresses specifically at the valves rather than the girdle bands, because the only discernable morphological change in the knockout mutant occurred in the valves.

A series of load-controlled nanoindentation experiments were conducted on five cell walls each from wild type and the mutant knockout-1. Wild-type cell walls exhibited an elastic deformation regime until a penetration depth of ~150 nm was reached

 3

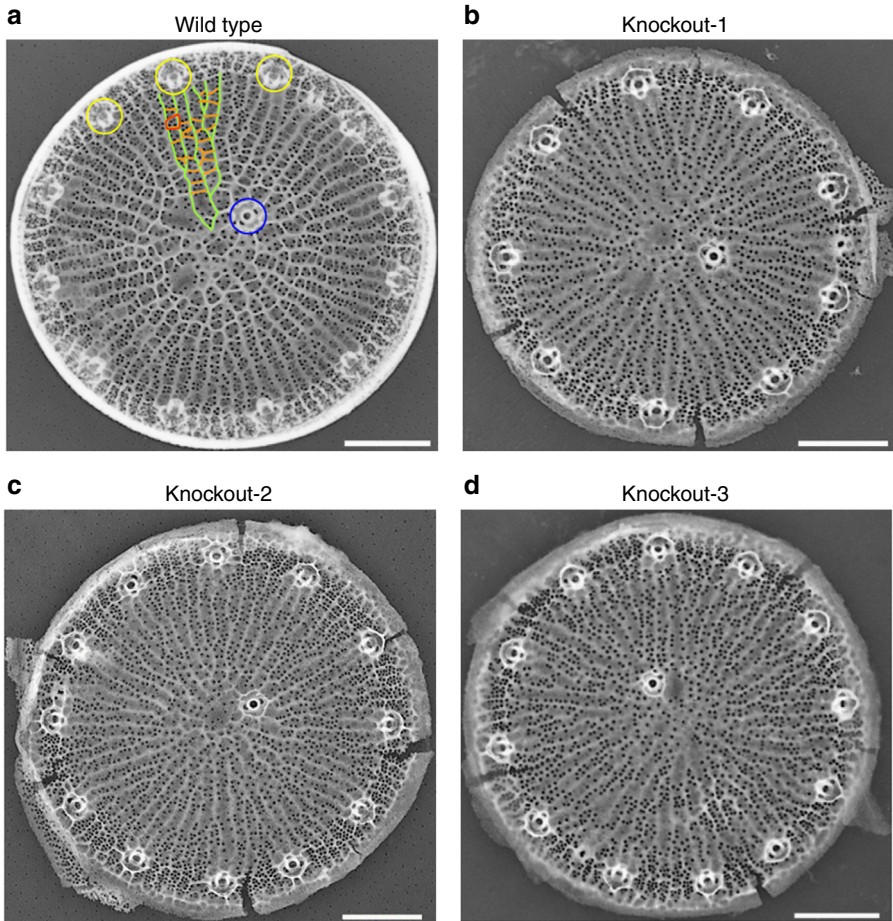

**Fig. 2** Comparison of biosilica morphology. Representative TEM images of *T. pseudonana* valve biosilica from **a** wild type and **b–d** the three Sin1 knockout clones. Due to the inverse contrast, silica has light gray or white color and pores through the silica and the background appear dark gray. Green line = costa, orange line = cross-connection, red circle = areola pore, yellow or blue circle = fultoportula. Note that cross-connections are largely absent in the knockout mutants. Scale bars: 1 μm

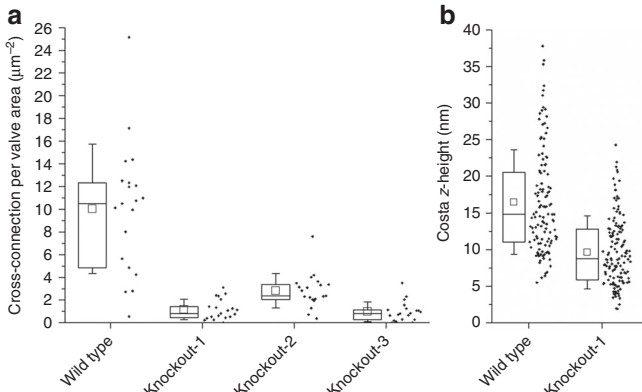

**Fig. 3** Quantitative analyses (box plots) of the valve biosilica structures from wild-type and knockout mutants. Whisker ends indicate upper and lower quartile; the mean is shown as a void square and the median is the horizontal line in the box. Each individual data point is to the right of the box plot. **a** Density of cross-connections. For each strain TEM images of 20 valves from different cells were analyzed. **b** Height of costae. AFM images from seven valves were analyzed for each wild type (120 data points) and mutant knockout-1 (139 data points)

(gray curves in Fig. 4g and Supplementary Fig. 9). After yielding, at forces between 4–5 μN, the cell walls either continued to deform elasto plastically or failed while still resisting the penetration of the tip (black curves in Supplementary Fig. 9). In contrast, the cell walls of the mutant knockout-1 were unable to resist the penetration of the indenter tip and the biosilica already failed when only comparatively low forces between the tip and the valve had develop (red curves in Supplementary Fig. 9). However, also here an elastic regime up to a penetration depth of ~150 nm was registered (red curves in Fig. 4g). The contact stiffness of the indenter tip-sample configuration was calculated based on the slope of the elastic regime, which was 17.61 ± 1.22 N m$^{-1}$ for the wild type and 4.93 ± 3.48 N m$^{-1}$ for the mutant knockout-1. These results indicated that the cell walls of the *Sin1* knockout mutant had a reduced mechanical strength and stiffness compared to the wild type. Movies recorded during displacement controlled nanoindentation up to a penetration depth of 1 μm clearly show that the valve of the wild-type cell wall completely returned to its initial shape after the diamond tip was retracted (Supplementary Movie 1, Fig. 4a, c). In contrast, the valve of the mutant knockout-1 buckled almost immediately after the contact with the tip that incurred an irreversible deformation (Supplementary Movie 2, Fig. 4d–f).

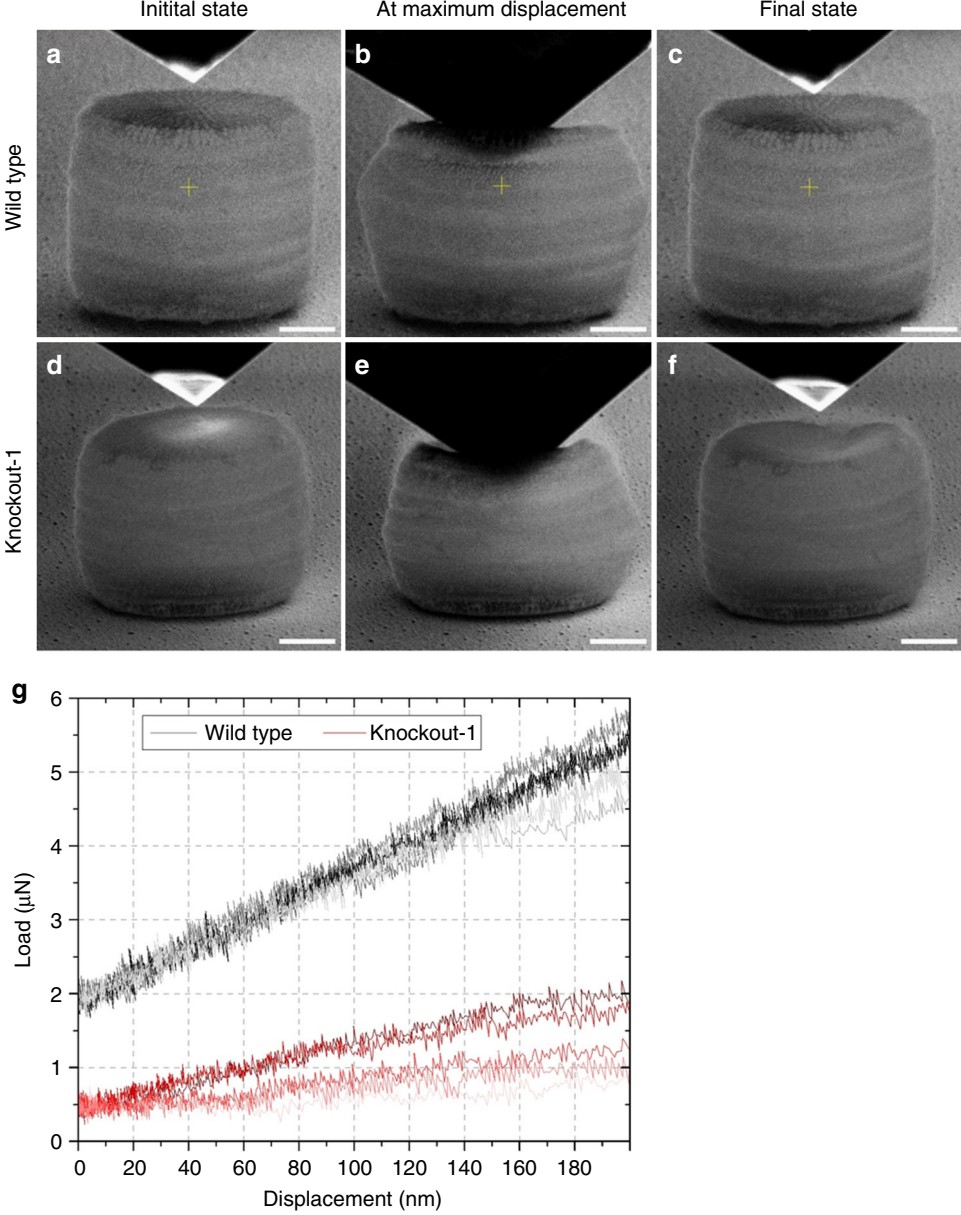

**Fig. 4** Mechanical testing of isolated *T. pseudonana* cell walls by nanoindentation. **a–f** SEM images extracted from the movies that were recorded during the displacement controlled nanoindentation experiments of a single cell wall from wild type and mutant knockout-1. The images compare the state of the cell walls from both specimens before, during, and after completion of the indentation experiment. The nanoindenter tip is above the specimen (left and right panels) or compressing the cell wall (middle panels). **g** Load–displacement curves for wild type (gray and black curves) and mutant knockout-1 (red curves) during load-controlled nanoindentation

## Discussion

The present work reports, to our knowledge, for the first time an alteration of a biologically produced silica pattern that has been achieved through a gene knockout. We have demonstrated here that the complete absence of the Sin1 protein leads to a phenotype with reduced capability to produce biosilica, a lack of hierarchical pore patterns, and impaired mechanical performance. By analyzing three independently generated knockout mutants and confirming the integrity of potential off-target sites, we essentially ruled out that the phenotype was caused by unintended mutations in the *T. pseudonana* genome. Our results reveal that the Sin1 protein is required for biosilica production in vivo but not essential. This is consistent with previous in vitro work showing that a recombinant Sin1 protein enhanced the silica formation activity of long-chain polyamines[18], but in this function can be

replaced by other abundant biosilica-associated proteins (e.g., silaffins, silacidins)[26,27]. The reduced *z*-height of costae but unaltered 2D patterns of costae and cribrum plates in the knockout mutants indicated that Sin1 is mainly involved in biosilica morphogenesis during the stage of *z*-axis expansion[25]. How Sin1 controls these defined morphogenic events cannot be deduced from the results of the current study. It will require the identification of Sin1 interaction partners, their properties, as well as the high-resolution localization of Sin1-bearing complexes during silica morphogenesis. During z-axis expansion, also the cross-linking of costae by biosilica bridges may occur, which is drastically reduced in the *Sin1* knockout mutants. The residual small number of cross-links in the knockout mutants may be due to genetic redundancy. The *T. pseudonana* genome encodes the Sin2 protein (the *Sin2* gene was intact in mutant knockout-1,

see Supplementary Fig. 10), which shares 55% sequence identity and the same domain structure with Sin1[18], suggesting similar functions. The *Sin2* gene is synchronously expressed with *Sin1*, but its mRNA is 60-fold less abundant[11]. A correspondingly low amount of the Sin2 protein in the knockout mutants would probably be insufficient to fully compensate for the loss of *Sin1*, thus giving rise to the observed phenotype.

It is highly likely that the presence of cross-connections will substantially contribute to the mechanical strength of the valve. If so, not only the reduced silica content but also the morphological changes in the biosilica would be responsible for the decreased cell wall stiffness of *Sin1* knockout mutants. Under the selective pressure of grazers and parasites in the natural habitat, a drastically weakened cell wall would likely be of considerable disadvantage regarding a diatom's fitness for survival[28]. The remarkable conservation of the *Sin1* gene in all groups of diatoms[18] may thus be a direct consequence of its essential function in the biosynthesis of mechanically robust diatom cell walls. We therefore hypothesize that maximizing mechanical stability is an important boundary condition for the morphogenesis of diatom silica patterns. This insight should be incorporated in future attempts to model this process. Furthermore, by revealing *Sin1* as a key player, we have paved the way for the genetic engineering of silica materials with designed mechanical stabilities. Such work will require an understanding of the role of Sin1 protein in the morphogenesis of diatom biosilica architectures that are quite different from that of *T. pseudonana*.

## Methods

**Culture conditions**. *T. pseudonana* (clone CCMP 1335) was grown at 18 °C with a 16 h light/8 h dark cycle at 5000–10,000 lux, in artificial seawater medium (NEPC) according to the North East Pacific Culture Collection protocol (http://www3. botany.ubc.ca/cccm/NEPCC/esaw.html), if not otherwise stated. For experiments carried out in $NH_4^+$ containing media, NEPC was prepared without $NaNO_3$ and autoclaved. Sterile filtered $NH_4Cl$ was added to a final concentration of 550 μM.

**Construction of the knockout plasmid pSin1KO**. The oligonucleotide primers used for construction of the plasmid are listed in the table below. The plasmid pTpNR-GFP/fcpNat(-KpnI)[29] was used as a starting point to construct pSin1KO. The first step comprised the removal of the *Bbs*I site in the *nat1* gene by inverse PCR and self-ligation using primer BbsI_removal_for and BbsI_removal_rev, which resulted in plasmid pTpNR-GFP/fcpNat(-KpnI/-BbsI). The *Cas9* gene was amplified from plasmid pSpCas9(BB)-2A-GFP[30] with primer Cas_for and Cas9_rev and cloned by In-Fusion® (Takara) into *EcoR*V and *Kpn*I digested pTpNR-GFP/fcpNat(-KpnI/BbsI), which resulted in plasmid pTpNR-Cas9-GFP/ fcpNAT. The *T. pseudonana* U6 Promoter was amplified from genomic DNA using primer U6_for and U6_rev. The gRNA scaffold was amplified from plasmid pSpCas9(BB)-2A-GFP using primer gRNA_for and tracrRNA_rev. Purified PCR products of both reactions were used as templates for gene SOEing PCR using primer U6_for and traceRNA_rev. The final PCR product was cloned by In-

Fusion® (Takara) into *Spe*I and *Xba*I digested pTpNR-Cas9-GFP/fcpNAT, which resulted in plasmid pTpNR-Cas9-GFP/fcpNAT/U6-BbsI.

*Sin1* specific gRNAs were introduced as described in the following. A solution of 1 μl of 100 ng μl$^{-1}$ of plasmid pTpNR-Cas9-GFP/fcpNAT/U6-BbsI was mixed with 0.5 μl cutsmart buffer (NEB), 0.5 μl T4 Ligase buffer (NEB), and 2 U *Bbs*I (NEB) in 9.0 μl total volume for 30 min at 37 °C. Separately, primers Sin1_gRNA-1/ 2_for and Sin1_gRNA-1/2_rev were mixed in 100 μl $H_2O$ at a final concentration 500 fmol μl$^{-1}$ each. The primer mix was heated to 95 °C and then the temperature was decreased to 40 °C in 5 °C increments (after each drop the temperature was maintained for 10 s). A 0.5 μl aliquot of the annealed primer mix and 200 U of T4 DNA ligase were (NEB) added to the *Bbs*I plasmid digest and incubated further for 30 min at 37 °C. An aliquot of 2.5 μl was used for transformation resulting in plasmids pTpNR-Cas9-GFP/fcpNAT/U6-Sin1_gRNA-1 and -2, respectively. Plasmid pTpNR-Cas9-GFP/fcpNAT/U6-Sin1_gRNA-2 was used as a template to amplify U6-Sin1_gRNA-2 with primer 2nd_U6_for and tracrRNA_rev. PCR product was cloned by In-Fusion® (Takara) into pTpNR-Cas9-GFP/fcpNAT/U6-Sin1_gRNA-1, digested with *Xba*I only, resulting in plasmid pSin1KO (Supplementary Fig. 1). The integrity of the plasmid was confirmed by test digestion and sequencing. Table 2 lists the primers that were used for PCR.

**Biolistic transformation**. The plasmid pSin1KO was introduced into *T. pseudonana* (grown on $NH_4^+$ containing NEPC medium) using the biolistic PDS-1000/He particle delivery system (Bio-Rad) according to the previously published method[21]. Briefly, 3 mg M17 tungsten particles were coated with 5 mg of circular plasmid DNA using the $CaCl_2$-spermidine method according to the manufacturer's instructions. A total of $10^8$ cells from a non-stationary culture of *T. pseudonana* (~$10^6$ cells mL$^{-1}$) were plated into the center (5 cm diameter) of an NEPC agar plate. After air drying, the plate was positioned into the biolistic chamber at a distance of 7 cm from the stopping screen. The cells were then bombarded with 600 ng of DNA-coated tungsten particles using 1100 psi rupture disks. After bombardment, the cells were scraped from plates, resuspended in $NH_4^+$ containing NEPC liquid medium (final density $10^6$ cells mL$^{-1}$), and incubated for 24 h in constant light. A total of $5 \times 10^6$ cells were plated onto NEPC agar plates (1.5% agar) containing 550 μM $NH_4Cl$ as sole N-source and 1.5% agar, supplemented with 150 μg mL$^{-1}$ nourseothricin (Jena Bioscience). For replating of clones, 5000 cells were transferred into 300 μl NEPC and spread onto a plate containing NEPC in 1.5% agar without antibiotic. Plates were incubated under constant light condition.

**PCR amplification of the *Sin1* and *Sin2* gene**. The DNA sequences of *Sin1* and *Sin2* are present in the Uniprot database under thapsdraft_24710 and thapsdraft_6586, respectively. The genes were amplified by colony PCR using Phusion High-Fidelity DNA Polymerase (Thermo Fisher) using the appropriate forward and reverse primer. For amplification 20 μL of complete PCR solution was prepared, and 0.25 μL of diatom cell culture was added. For initial primer denaturation and cell lysis, the solution was incubated at 98 °C for 3 min. The PCR products were gel purified and sequenced using the sequencing primers Sin1_seq, Sin2_seq_for, and Sin2_seq_rev (see table below). Table 3 lists the primers that were used for PCR amplification and sequencing.

**PCR amplification of potential off-target sites for gRNA_1**. Off-target 1 was predicted to be on chromosome 1 within the gene thapsdraft_20571, off-target 2 on chromosome 5 within the gene thapsdraft_22527, off-target 3 on chromosome 10 within the gene thapsdraft_29506, and off-target 4 on chromosome 16b within the gene thapsdraft_25190. For each position an ~1000 bp fragment that covers the potential off-target region was amplified, and the PCR products were sequenced using the corresponding reverse primers (see table below). Table 4 lists the primers that were used for PCR amplification and sequencing.

### Table 2 PCR primers that were used in construction of the knockout plasmid pSin1KO

| Primer | Sequence 5′ → 3′ (*Sin1* specific gRNA sequences are underlined) |
| --- | --- |
| BbsI_removal_for | P-CGCGTCACCGCCACCGG |
| BbsI_removal_rev | P-AAAAACGGTGTCGGTGGTGAAGG |
| Cas9_for | AACACAAAGGAACCAACAGATATCATGGCCCCAAAGAAGAAGCGGAAG |
| Cas9_rev | GCTCACCATTCCGCCGGTACCGTCGCCTCCAAGTTGAGACAG |
| U6_for | TCTCCCGGGGGATCCACTAGTTCTTCATCAAGAGAGCAACC |
| U6_rev | GAAGACCCAATTTCGGCAAAACGTTAATTTATAAAGC |
| gRNA_for | GCCGAAATTGGGTCTTCGAGAAGACCTGTTTTAG |
| tracrRNA_rev | GGTGGCGGCCGGCAGCTCTAGAGCAAAAAAAGCACCGACTCG |
| Sin1_gRNA-1_for | AATTGTGCAATCCTCCTCGCCACCGCCA |
| Sin1_gRNA-1_rev | AAACTGGCGGTGGCGAGGAGGATTGCAC |
| Sin1_gRNA-2_for | AATTGTGCCAGCAGTGCGTAGAGATGC |
| Sin1_gRNA-2_rev | AAACGCATCTCTACGCACTGCTGGCAC |
| 2nd_U6_for | GGTGCTTTTTTTGTTCTAGCTTCATCAAGAGAGCAACC |

### Table 3 Primers that were used for PCR amplification and sequencing of Sin1 and Sin2 genes

| Primer | Sequence 5′ → 3′ |
| --- | --- |
| Sin1_for | ATCAACGAGAAGATGCACG |
| Sin1_rev | AGACTACGACACTCCCTCC |
| Sin1_seq | ACGCCTGGTGCTACATGAC |
| Sin2_for | CGATGATGGCAGTGGCTTGAG |
| Sin2_(seq)_rev | CAGCCGACGAGGTTTCCACTAATTG |
| Sin2_seq_for | CAGCGCATCGCAACGTACCCC |

### Table 4 Primers that were used for PCR amplification and sequencing of potential off-target sites

| Primer | Sequence 5′ → 3′ |
| --- | --- |
| 20571_OT1_for | CACGTTGTTCATGTCCACCG 3′ |
| 20571_OT1_rev | GATTGGCGGCACACCCC |
| 22527_OT2_for | TTGCTGTGCTGCTGCTATCC |
| 22527_OT2_rev | AAATTGCATCTTCAACACCAGTC |
| 29506_OT3_for | AACGTTGGTATTGTGAATTGTATC |
| 29506_OT3_rev | CATATTGATACGTTGCATTCTTTG |
| 25190_OT4_for | CTCATCTTCATGACGTATCCG |
| 25190_OT4_rev | ATCTCATTTGCAAAACATTGTGG |

**Western blot**. To enhance expression of the Sin1 protein, the cell culture was synchronized by the silicon starvation-replenishment method according to a previously published protocol[31]. Briefly, cell culturing was performed in 100 mL ASW medium[32]. When cell density reached $5 \times 10^5$ cells mL$^{-1}$, the culture was washed twice with 40 mL Si-free ASW and resuspended in 100 mL Si-free ASW for ~24 h under aeration in polycarbonate flasks. Subsequently, Na$_2$SiO$_3$ was added to the culture to a final concentration of 210 μM, and after 3 h of cultivation cells were centrifuged (3220 g, 10 min, and 18 °C). The pellet was resuspended in 1 mL of NEPC medium, 300 μL of glass beads were added and vortexed for $3 \times 30$ s. To 1 mL of the cell lysate 250 μL of 5× SDS-loading buffer (250 mM Tris/HCl pH 6.8, 10% (w/v) SDS, 30% (v/v) glycerol, 62.5 mM EDTA pH 8.0, 5% (v/v) β-mercaptoethanol, and 0.05% (w/v) bromophenol blue) was added to the sample and incubated at 95 °C for 10 min. An appropriate volume containing $5 \times 10^5$ cells was loaded onto each lane of a TGX Stain-Free 4–20% SDS-PAGE (Bio-Rad). After electrophoresis, the SDS-PAGE was blotted onto a 0.2-μm PVDF mini membrane using the Trans-Blot Turbo Blotting System (Bio-Rad). The membrane was incubated with 1× Roti®-Block (Carl Roth) for 60 min at room temperature, and then washed once with PBST (137 mM NaCl, 2.7 mM KCl, 10 mM Na$_2$HPO$_4$, 1.8 mM KH$_2$PO$_4$, and 0.05% (v/v) Tween-20). The blot was simultaneously incubated with anti-rSin1$^{lum}$ antiserum[18] (1:1000 in PBST) and anti-Histone-3 antibodies (Agrisera; 1:20,000 in PBST). Following 60 min incubation at room temperature, the membrane was washed three times for 10 min with PBST and subsequently incubated with goat anti-rabbit IgG peroxidase conjugate (Sigma-Aldrich; 1:10,000 in PBST). After 60 min incubation at room temperature, the membrane was washed twice for 10 min in 20 mL PBST and twice in 20 mL PBS. The membrane was transferred into a disposal bag before incubating with 2 mL SuperSignal West Pico chemiluminescent substrate (Thermo Scientific) for 5 min at room temperature. Chemiluminescence was detected using the ChemiDoc MP imaging system (Bio-Rad) with an exposure time of 1 s.

**Determining cell growth rates, silica content, and size**. Cells were grown in 300 mL NEPC medium and each strain was subjected to three consecutive growth cycles (inoculation at ~3000 cells mL$^{-1}$, growth to ~$10^6$ cells mL$^{-1}$, and then dilution down to ~3000 cells mL$^{-1}$). Cell densities were measured in the fourth growth cycle with at least two measurements being taken at each time point for each strain using the TC20 Automated Cell Counter (Bio-Rad). To calculate the growth rates, time points between 7 and 16 days were fitted by linear regression.

To determine the total silica content 13 mL of ~$10^6$ cells mL$^{-1}$ (exact cell number was determined directly from the culture) were centrifuged (3220× g, 10 min, 18 °C). After careful removal of the supernatant, the pellet was washed once with 13 mL of H$_2$O (3220 × g, 10 min, and 18 °C), and finally resuspended in 1 mL H$_2$O. Subsequently, 500 μL of 6 M NaOH was added and incubated at 95 °C for 30 min. A 50 μL aliquot was removed to determine the silicic acid content according to a previously published method[33]. Briefly, to 50 μL sample the following solutions were added with mixing after each addition: 150 μL H$_2$O, 100 μL acetic acid, 40 μL sulfate buffer (1.25 M NaHSO$_4$, 0.75 M Na$_2$SO$_4$), and 20 μL molybdate solution (80 mM (NH$_4$)$_6$Mo$_7$O$_{24}$). For all subsequent incubation steps the sample was left in the dark. After incubation for 10 min at 60 °C, the solution was kept for 5 min at room

temperature, then 100 μL citrate solution (2.2 M citric acid in 0.4 M HCl) was added, and the mixture incubated for 20 min at room temperature. Finally, 10 μL reducing solution (100 mM 4-amino-3-hydroxy-1-naphthalenesulfonic acid, 95 mM Na$_2$SO$_3$, and 63 mM Na$_2$S$_2$O$_5$) was added, the mixture incubated for 20 min at room temperature, and the absorption of the solution was determined at $\lambda = 800$ nm. Sodium metasilicate pentahydrate was used as a standard.

To determine the cell size, bright-field light microscopy images were taken prior to silica content measurements using a Zeiss Axiovert 200 instrument at ×500 magnification. Cell sizes were measured using the Fiji software.

**Preparation of biosilica for electron microscopy**. For biosilica isolation 100 mL of ~$10^6$ cells mL$^{-1}$ were harvested (3220 × g, 10 min). The pellet was resuspended in 6 mL extraction buffer (2% SDS, 100 mM EDTA pH 8, and 1 mM PMSF), and continuously shaken at 55° for 1 hour to solubilize intracellular material. The biosilica was pelleted (3220 × g, 2 min) and washed twice in 2 mL 10 mM EDTA pH 8, and 1 mM PMSF. Extraction and washing were repeated until the biosilica was colorless. The colorless biosilica was washed with 2 mL acetone, four times with 5 mL H$_2$O, and finally resuspended in 1 mL H$_2$O.

For scanning electron microscopy intact biosilica was stepwise exchanged from H$_2$O to 100% ethanol by washing (3220 g, 2 min) with incrementally increasing ethanol concentrations (20%, 40%, 60%, 80%, and 100%). The water-free biosilica was critical point dried using the Leica CPD 300 instrument (Leica Microsystems). The dried biosilica was mounted on carbon pads on aluminum stubs and sputter coated with platinum using a Baltec SCD 050 instrument and argon as the process gas (40 mA, 40 s). Secondary electron microscopy images were taken using a JSM 7500F field emission scanning electron microscope (Jeol) at an acceleration voltage of 5 kV.

To detach girdle bands from valves, isolated biosilica was resuspended in 500 μL H$_2$O in a 1.5-mL tube and sonicated with an MS72 sonotrode tip (Bandelin) applying a total of 0.12 kJ over 5 s. For transmission electron microscopy (TEM) 10 μL of sonicated biosilica was transferred onto Formvar-coated gold-coated finder grids (G200F2-Au from EMS) that were strengthened with evaporated carbon. The majority of the liquid was removed by blotting with a piece of filter paper, washed four times with 50 μL H$_2$O and samples were air-dried over-night before imaging. TEM images were taken using either a Morgagni 268D (FEI) instrument or a Jeol JEM-1400 TEM instrument at an acceleration voltage of 80 kV.

**Atomic force microscopy (AFM)**. Isolated biosilica was immobilized on a poly-L-lysine-coated glass specimen holder by incubation for 1 h at room temperature. The AFM measurements were carried out on a Nanowizard IV (JPK Instruments) utilizing the QI$^{TM}$ measurement mode with a trigger force of 200 pN. Biolever Mini (BL-AC40-TS, Olympus Micro) was used as cantilevers for the measurements. Calibration of the cantilevers was done by the contact-free method according to the manufacturer's instructions. Image correction and flattening were performed within the JPK data processing software (JPK Instruments). The height of the costae was calculated using the "Extract profile" tool of the program Gwyddion. For this purpose, a line of at least 100 nm was drawn on top of a costa and in the valley next to this costa to calculate their average height profile. The net height of a costa was calculated by subtracting the average height of the valley from the average height of its neighboring costa.

**Nanoindentation**. Biosilica isolated from *T. pseudonana* cells was critical point dried using the Leica CPD 300 instrument (Leica Microsystems) and transferred onto palladium-coated glass cover slips (0.5 cm diameter) attached to aluminum stubs and fixed with a conductive glue. The stubs were mounted in an in situ picoindentation system PI 85 (Bruker) equipped with a cube-corner diamond tip installed inside the Scios DualBeam electron microscope (Thermo Fisher). During the experiments, load-displacement curves and live footage of the indentation procedure were recorded.

By enforcing contact between the tip and the center of the valves the indentation experiments were performed in two modes: load controlled and displacement controlled. In the load controlled mode, a load function of 10 s loading up to a maximal force of 10 μN, 2 s holding time followed by 2 s unloading time was used. In the case of wild-type biosilica, an initial contact of 2 μN between the tip and the valves was established before performing the experiment. In the case of mutant knockout-1, due to high compliance of biosilica, an initial contact of only 0.5 μN was established. The contact stiffness of the indenter tip-sample configuration was measured by calculating the slope of the linear part of the load–displacement curves, at an interval between 0 and 150 nm, using the least squares method. At least five indentation experiments per cell wall type were performed. In the displacement controlled mode, the tip was lowered into the cell wall until a penetration of 1 micron was achieved. The load function consisted of 10 s loading time, and 10 s unloading time.

**Statistics and reproducibility**. Statistical analysis was carried out using the software packages Excel and OriginPro. Average and standard deviation were calculated for measurements of silica content, cell size, cell density, and growth rate. The growth rate was calculated within the linear growth regime by linear least-squares fitting using data points within the cell density range from $5 \times 10^5$ cells mL$^{-1}$ to $2 \times 10^5$ cells mL$^{-1}$. The density of cross-connections and the costa z-height in the valves were calculated by the median of the datasets according to the statistical variance of the data points. No data points were excluded from the calculations.

Measurements for silica content and growth rate were carried out at least three times, cell counts at least twice, and cell size once. The contact stiffness of the diamond tip-sample configuration for each indentation experiment was calculated as the slope of the initial part of the load–displacement curve, at an interval between 0–150 nm, using the linear least squares fitting in OriginPro. The final value of the contact stiffness is expressed as an average of five nanoindentation experiments ± standard deviation.

**Reporting summary**. Further information on research design is available in the Nature Research Reporting Summary linked to this article.

## Data availability

The authors declare that all data supporting the findings of this study are available within the article and its supplementary information files or are available from the corresponding author upon request. The source data for Figs. 3 and 4 and for relevant tables and supplementary items are available as Supplementary Data 1–5.

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

## Acknowledgements

We would like to thank Thomas Kurth (CMCB Technology Platform, TU Dresden), Elke Reich, and Deborah Stier (B CUBE, TU Dresden) for help with electron microscopy, Jennifer Klemm (B CUBE, TU Dresden) for help with diatom cultivation and DNA cloning, and Nicole Poulsen (B CUBE, TU Dresden) for critically reading the manuscript. We are indebted to Frank Buchholz (Faculty of Medicine, TU Dresden) for providing plasmid pSpCas9(BB)-2A-GFP (Addgene #48138). This work was supported by the Deutsche Forschungsgemeinschaft (DFG) through Research Unit 2038 "NANOMEE" (KR 1853/8-2) to N.K. and by the Bundesministerium für Bildung und Forschung (BMBF) through grant 03Z22EN11 to I.Z.

## Author contributions

N.K. conceived the project; N.K., S.G., D.P. and I.Z. designed experiments; S.G., D.P., and I.Z. performed experiments and analysed data; N.K. and S.G. wrote the paper with input from I.Z.

## Additional information

**Competing interests:** The authors declare no competing interests.

