## [Peer Review File · Communications Biology]

Reviewers' comments:

Reviewer #1 (Remarks to the Author):

The paper describes the knock-out of the Silicanin-1 (Sin-1) gene in *T. pseudonana*. The authors provide clear evidence from a genetic and proteomic perspective to show that the Sin-1 gene is no longer active. Physiological and morphological analyses show that the gene is involved in governing silica content and structural integrity. Experiments appear to have been carried out robustly, with sufficient replicates and data points to support their findings. The manuscript is clear and concise, as are the figures. The research adds new knowledge to the field of biomineralization and diatom biology.

Minor comments

Page 2 line 2. I would suggest making this first sentence a little less finite - all known diatoms produce a silica based shell (at least for one morphotype of the sp.), as the described number of species (~15,000) is lower than the expected number of species (>100,000) we can't know for sure that there are no diatoms out there with no silica frustule (possible as certain morphotypes of *P. tricornutum* do not require silicon to grow).

Page 2 last line. The promoter has been previously described in ref 20 – has the sequence of the promoter come from this source or has it been independently determined. If it is the former, then it needs to be referenced.

Page 3 line. Typo – change suggested to suggested.

Page 3 line 8. Mosaic colonies with a mixture of WT and mutant genes following CRISPR have been previously reported in diatoms (ref 20 and DOI, 10.1038/srep24951). This could be worth mentioning in support of the presented data.

Page 3 line 22. The line 'For simplicity sub-clones C4... is very useful for quickly understanding supplementary Fig. 3A. If this line was moved higher in the text before or shortly after the first reference to supplementary Fig. 3A it would make the figure clearer, especially since sub-clones are not labelled in the order they are mentioned in the text.

Page 8, last line. Suggest changing '...biosilica are responsible for...' to '... biosilica are likely to be responsible for...', in line with the previous sentence.

Page 9 line 4. This sentence appears to be missing its reference (18).

Page 9 line 9. Typo – change wioth to with.

Page 9 2nd to last line. Typo – change seperatly to separately .

Page 10 line 6. Typo – change templete to template.

Page 10 Biolistic transformation paragraph. Some explanation why the cells are initially grown in NH₄⁺ media and then changed to NO₃ media is needed. Presumably since the Cas9 is under the control of an NR promoter this should ensure the Cas9 is not active until it is re-plated on NO₃ plates. Why was this route chosen?

Page 11 line 15. Typo – change ‘...synchronized by silicon the starvation...’ to ...‘synchronized by the silicon starvation...’.

Page 12 line 32. Typo – change resuspenden to resuspended.

Page 12 line 33. Typo – One of the units needs to be deleted from ‘30 min hour’.

Page 12 2nd to last line. A brief description of the method is needed, especially as the current paper is written in English and the reference is in German, which makes it difficult to recreate the method for the readers. Alternatively an additional reference, which describes the method in English could be added.

Page 13 line 1. Typo – change ‘...were taken prior silica...’ to ‘...were taken prior to silica...’.

Page 13 line 34. Typo – change calculate to calculated.

Page 14 2nd to last line. Typo – change second to seconds.

Supplementary Page 3 line 8. Typo – change ‘...plasmids were from six E. coli...’ to ‘...plasmids from six E. coli...’.

Supplementary Page 7 line 2. Pluralisation needs to be addressed in this sentence. Suggest changing to ‘Red lines indicate costae and each green line indicates the position...’.

Supplementary Page 9 line 2. Typo – change target to targets.

Supplementary Page 9 line 3. Typo – change silica contents to silica content.

Reviewer #2 (Remarks to the Author):

The manuscript by Görlich et al. describes the mutant phenotype resulting from inactivation (via CRISPR-Cas9) of a protein localized to the silica deposition vesical in the diatom *Thalassiosira pseudonana*. The authors carefully document that the knock-out mutants contain significantly less biosilica and have resulting defects in the structure of the silica cell wall. They convincingly show that these changes compromise the strength and stiffness of the cell walls. Importantly, the authors show that the resulting phenotype did not result from other off-target loci. Experimentally, this is beautiful work, with use of appropriate statistics. Their study advances our understanding of the somewhat mysterious generation of remarkably diverse and intricate silica-based diatom cells walls.

My only suggestions have to do with further explaining the implications of their results. For example, the authors reference Kotzsch, A. et al. (BMC Biol. 15, 9–11, 2017). The Kotzsch et al manuscript provides hypotheses as to the role of Sin1 in silica precipitation. It would be helpful if the authors included these hypotheses in this manuscript and explicitly described how the in vivo results support (or modify) their previous hypotheses. Second, the authors conclude the abstract and the discussion with the statement that their result “paves the way for genetic engineering of silica architectures with desired structures and mechanical performance.” It is not clear how their results actually do this – the authors could be more explicit here. Finally, the authors should also explain what they mean when

they state that their result “highlights that understanding biosilica morphogenesis will also require fundamental insight into the mechanical constraints for cell wall stability.”

Reviewer #3 (Remarks to the Author):

The authors have investigated the role of silicanin-1 (Sin1) –a protein hypothesized to play a key role in biosilica formation of diatoms– on the morphology and mechanical properties of *T. pseudonana*. The approach is innovative and novel in the field of biomineralization, exploiting CRISP-Cas9 to knock-out the gene encoding Sin1.

The inactivation of the Sin1 gene appears to have been carefully executed and the verification of successful knock-out, which is a central aspect of the paper, is convincing. The authors have then conducted morphological characterization and nanomechanical testing to demonstrate that Sin1 plays an important role in the structural integrity of the diatoms wall.

I only have a few comments about the study, mostly regarding the nanomechanical characterization:

- The authors have used a sharp tip to deform individual valves of isolated cell calls. With such a geometry, the contact stresses are concentrated at a point rather than distributed along the entire cell wall. Why not using a flat cylindrical punch to conduct full “nano-compression” tests, in which valves would be fully compressed until failure ? It seems to me these tests would provide even more convincing evidence about the difference in mechanical properties of the wild type vs. the KO mutant. Actually since the diameter of the valve is around 3µm (see Fig. 4), even a spherical tip geometry with a spherical radius of 15µm or above would generate a similar compressive field, and would thus be sufficient to compress entire valves.

- In Page 8, using in situ videos of nanoindentation testing, the authors are mentioning that the wild type cell wall is returning to its initial shape whereas for the KO1 mutant, there is irreversible deformation. The videos support this statements but I did not find it that obvious. I think what would be useful to make the claim fully convincing would be to draw the contour of the valves both at the beginning and at the end of the tests, and to then demonstrate that the KO1 mutant is indeed irreversibly deformed. Any image processing software should work to draw the contours.

- A very minor point: for Supplementary Fig. 7, the caption should mention that these are AFM images. This is mentioned in the text but not in the caption.

Response to the reviewers' comments

Reviewer comments are shown in italics and our responses in bold font.

Reviewer #1

*Page 2 line 2. I would suggest making this first sentence a little less finite - all known diatoms produce a silica based shell (at least for one morphotype of the sp.), as the described number of species (~15,000) is lower than the expected number of species (>100,000) we can't know for sure that there are no diatoms out there with no silica frustule (possible as certain morphotypes of *P. tricornutum* do not require silicon to grow).*

Diatoms are a large group of single-celled microalgae (~15,000 described and >100,000 estimated species) renown for their silica-based cell walls that exhibit species specifically ornamented morphologies with patterns of nano- to micrometer sized pores¹.

Page 2 last line. The promoter has been previously described in ref 20 – has the sequence of the promoter come from this source or has it been independently determined. If it is the former, then it needs to be referenced.

Reference 20 has been included in the sentence.

Page 3 line. Typo – change sugested to suggested.

Done

Page 3 line 8. Mosaic colonies with a mixture of WT and mutant genes following CRISPR have been previously reported in diatoms (ref 20 and DOI, 10.1038/srep24951). This could be worth mentioning in support of the presented data.

A sentence has been added that includes both references: “Mosaic colonies have previously been observed in diatoms that were subjected to CRISPR/Cas9 based genome editing^{20, 33}.”

Page 3 line 22. The line ‘For simplicity sub-clones C4... is very useful for quickly understanding supplementary Fig. 3A. If this line was moved higher in the text before or shortly after the first reference to supplementary Fig. 3A it would make the figure clearer, especially since sub-clones are not labelled in the order they are mentioned in the text.

As suggested the information was moved up, right after the first sentence mentioning Supplement Figure 3A (page 3, line 15). The following sentence was updated to account for the re-naming.

Page 8, last line. Suggest changing ‘...biosilica are responsible for...’ to ‘... biosilica are likely to be responsible for...’, in line with the previous sentence.

To avoid the frequent mentioning of “likely”, which is used also in the preceding sentence, we have changed this section as follows (changes are underlined): “It is highly likely that the presence of cross-connections will significantly contribute to the mechanical strength of the valve. If so, not only the reduced silica content but also the

morphological changes in the biosilica would be responsible for the decreased cell wall stiffness of Sin1 knockout mutants.”

Page 9 line 4. This sentence appears to be missing its reference (18).

The reference has been added.

Page 9 line 9. Typo – change wioth to with. Done.

Page 9 2nd to last line. Typo – change seperatly to separately . Done.

Page 10 line 6. Typo – change templete to template. Done.

Page 10 Biolistic transformation paragraph. Some explanation why the cells are initially grown in NH₄⁺ media and then changed to NO₃ media is needed. Presumably since the Cas9 is under the control of an NR promoter this should ensure the Cas9 is not active until it is re-plated on NO₃ plates. Why was this route chosen?

The reviewer is correct that we initially intended to permanently switch off Cas9 after it the knockout mutation in the Sin1 gene had been introduced. However, realized that growing *T. pseudonana* wild type cells in NH₄⁺ bearing NEPC medium led to a minor but non-negligible fraction of cells with aberrant morphology. The morphological aberrations vanished completely after the cells were transferred into NO₃⁻ bearing NEPC medium and grown for 2 weeks. Since morphological stability of the wild type cells was an important reference point for the characterization of our knockout mutants, we abandoned the original plan of switching off Cas9 by growth on NH₄⁺ bearing medium. Consequently, in the present report only wild type and mutant cells were examined that were grown for >3 months in NO₃⁻ bearing medium.

We feel that including our observation of the still enigmatic effect of NH₄⁺ on morphology would be distracting rather than provide additional insight for the study described in this paper. Therefore, we would prefer to not explain why we permanently switched to NO₃⁻ medium after the cells were initially grown for a relatively brief period (1 week in liquid culture, 10 days on plates) in NH₄⁺ bearing NEPC medium.

Page 11 line 15. Typo – change ‘...synchronized by silicon the starvation...’ to ...’synchronized by the silicon starvation...’. Done.

Page 12 line 32. Typo – change resuspenden to resuspended. Done.

Page 12 line 33. Typo – One of the units needs to be deleted from ‘30 min hour’.

The time information was changed to “30 min”.

Page 12 2nd to last line. A brief description of the method is needed, especially as the current paper is written in English and the reference is in German, which makes it difficult to recreate

the method for the readers. Alternatively an additional reference, which describes the method in English could be added.

The following paragraph was added: “In the following, the method for silicic acid determination is briefly described. To 50 µl sample the following solutions were added with mixing after each addition: 150 µl H₂O, 100 µl acetic acid, 40 µl sulfate buffer (1.25 M NaHSO₄, 0.75 M Na₂SO₄), and 20 µl molybdate solution (80 mM (NH₄)₆Mo₇O₂₄). For all subsequent incubation steps the sample was left in the dark. After incubation for 10 min at 60°C, the solution was kept for 5 min at room temperature, then 100 µl citrate solution (2.2 M citric acid in 0.4 M HCl) was added, and the mixture incubated for 20 min at room temperature. Finally, 10 µl Reducing solution (100 mM 4-Amino-3-hydroxy-1-naphthalenesulfonic acid, 95 mM Na₂SO₃, 63 mM Na₂S₂O₅) was added, the mixture incubated for 20 min at room temperature, and the absorption of the solution was determined at λ = 800 nm.”

Page 13 line 1. Typo – change ‘...were taken prior silica...’ to ‘...were taken prior to silica...’.
Done.

Page 13 line 34. Typo – change calculate to calculated. **Done.**

Page 14 2nd to last line. Typo – change second to seconds. **Done.**

Supplementary Page 3 line 8. Typo – change ‘...plasmids were from six E. coli...’ to ‘...plasmids from six E. coli...’. **Done.**

Supplementary Page 7 line 2. Pluralisation needs to be addressed in this sentence. Suggest changing to ‘Red lines indicate costae and each green line indicates the position...’

The sentence now reads: “A red line indicates a costa and a green line indicates the position of a cribrum plate in the middle between two adjacent costae.”

Supplementary Page 9 line 2. Typo – change target to targets. **Done.**

Supplementary Page 9 line 3. Typo – change silica contents to silica content. **Done.**

Reviewer #2

My only suggestions have to do with further explaining the implications of their results. For example, the authors reference Kotzsch, A. et al. (BMC Biol. 15, 9–11, 2017). The Kotzsch et al manuscript provides hypotheses as to the role of Sin1 in silica precipitation. It would be helpful if the authors included these hypotheses in this manuscript and explicitly described how the in vivo results support (or modify) their previous hypotheses.

The reviewer seemed to have missed the paragraph in lines 7-13 of the Discussion section, which addresses the relationship between the previously published *in vitro* results on recombinant Sin1 and the new *in vivo* data that are presented here:

“Our results reveal that Sin1 is required for biosilica production *in vivo* but not essential. This is consistent with previous *in vitro* work showing that recombinant Sin1 enhances the silica formation activity of long-chain polyamines¹⁸, but in this function can be replaced by other abundant biosilica-associated proteins (e.g., silaffins, silacidins)^{25,26}. The reduced z-height of costae but unaltered 2D patterns of costae and cribrum plates in the knockout mutants indicated that Sin1 is mainly involved in biosilica morphogenesis during the stage of z-axis expansion²⁴.”

To highlight this paragraph a bit more, we added the following two sentences that point to the remaining open questions about Sin1:

“How Sin1 controls these defined morphogenic events cannot be deduced from the results of the current study. It will require the identification of Sin1 interaction partners, their properties as well as the high resolution localization of Sin1-bearing complexes during silica morphogenesis.”

Second, the authors conclude the abstract and the discussion with the statement that their result “paves the way for genetic engineering of silica architectures with desired structures and mechanical performance.” It is not clear how their results actually do this – the authors could be more explicit here.

We agree with the reviewer that this statement deserves further explanation. Therefore, we have added the following sentence at the end of the discussion section: “Such work will require an understanding of the role of Sin1 in the morphogenesis of diatom biosilica architectures that are quite different from that of *T. pseudonana*.”

Finally, the authors should also explain what they mean when they state that their result “highlights that understanding biosilica morphogenesis will also require fundamental insight into the mechanical constraints for cell wall stability.”

We have reworded and split this sentence in two, which should clarify this issue: “We therefore hypothesize that maximizing mechanical stability is an important boundary condition for the morphogenesis of diatom silica patterns. This insight should be incorporated in future attempts to model this process.”

Reviewer #3

- The authors have used a sharp tip to deform individual valves of isolated cell calls. With such a geometry, the contact stresses are concentrated at a point rather than distributed along the entire cell wall. Why not using a flat cylindrical punch to conduct full “nano-compression” tests, in which valves would be fully compressed until failure ? It seems to me these tests would provide even more convincing evidence about the difference in mechanical properties of the wild type vs. the KO mutant. Actually since the diameter of the valve is around 3um (see Fig. 4), even a spherical tip geometry with a spherical radius of 15um or above would generate a similar compressive field, and would thus be sufficient to compress entire valves.

We agree with the reviewer that by using a relatively sharp tip we concentrate the stresses at the valves of the tested biosilica. In fact, this experimental geometry was

chosen intentionally as the only discernable morphological change in the knockout mutant occurred in the valves. Moreover, as it can be seen in the videos recorded during the indentation experiments, this geometry probes the deformation behavior of the entire cell wall that includes both the valves and the girdle bands. In contrary, if we would use the experimental set-up suggested by the reviewer, we would only deform the girdle bands and thus, overlook the mechanical behavior of the valves. In contrast, the experimental set-up that we chose allowed probing both, the deformability of the valve and the overall mechanical properties of the biosilica cell wall, which was the purpose of our mechanical measurements.

- In Page 8, using in situ videos of nanoindentation testing, the authors are mentioning that the wild type cell wall is returning to its initial shape whereas for the KO1 mutant, there is irreversible deformation. The videos support this statements but I did not find it that obvious. I think what would be useful to make the claim fully convincing would be to draw the contour of the valves both at the beginning and at the end of the tests, and to then demonstrate that the KO1 mutant is indeed irreversibly deformed. Any image processing software should work to draw the contours.

We thank the reviewer for this excellent suggestion. We added Supplement Figure 9, which shows images extracted from the videos. They present the state of the cell walls of the wild type and knockout strain before, during and after the indentation experiment in a comparative manner (see below).

Supplement Figure 9. SEM images extracted from the movies that were recorded during the displacement controlled nanoindentation experiments of a single cell wall from wild type (movie S1) and mutant KO1 (movie S2). The images compare the state of the cell walls from both specimens before, during and after completion the indentation experiment.

- A very minor point: for Supplementary Fig. 7, the caption should mention that these are AFM images. This is mentioned in the text but not in the caption.

The following title has been added to the caption: "Atomic force microscopy imaging of biosilica valves."